# Novel Therapeutic Strategies Exploiting the Unique Properties of Neuroendocrine Neoplasms

**DOI:** 10.3390/cancers15204960

**Published:** 2023-10-12

**Authors:** Maryam Safari, Luigi Scotto, Thomas Litman, Lubov A. Petrukhin, Hu Zhu, Min Shen, Robert W. Robey, Matthew D. Hall, Tito Fojo, Susan E. Bates

**Affiliations:** 1Division of Hematology/Oncology, Department of Medicine, Columbia University Medical Center, New York, NY 10032, USA; 2Department of Immunology and Microbiology, University of Copenhagen, 1172 Copenhagen, Denmark; 3National Center for Advancing Translational Sciences (NCATS), National Institutes of Health, Rockville, MD 20892, USA; 4Developmental Therapeutics Branch, National Cancer Institute, National Institutes of Health, Bethesda, MD 20892, USA; 5James J. Peters Bronx Veterans Affairs Medical Center, Bronx, NY 10468, USA; 6Herbert Irving Comprehensive Cancer Center, Columbia University Medical Center, New York, NY 10032, USA

**Keywords:** neuroendocrine, neuroendocrine neoplasms, metabolism, YAP1, NAMPT inhibitors, HDAC inhibitors

## Abstract

**Simple Summary:**

The screening of experimental and repurposing drugs has enormous potential for rapid clinical impact, particularly when new therapeutic strategies are sought. By screening two drug libraries, nicotinamide phosphoribosyltransferase (NAMPT) and histone deacetylase (HDAC) inhibitors were identified as agents with the highest activity against neuroendocrine tumor-derived cell lines. An evaluation of different NAMPT inhibitors revealed a reduction in basal oxidative phosphorylation and energy production to be partially but not completely responsible for cell death following the exposure to the drug, with the yes-associated protein 1 (YAP1) playing a pivotal role in the sensitivity to the NAMPT inhibitors. Insight into the contribution of YAP1 guided the evaluation of combinations of HDAC and NAMPT inhibitors for the treatment of neuroendocrine neoplasms.

**Abstract:**

Background: Over the last few decades of treatment, the outcomes for at least some subsets of neuroendocrine neoplasms (NENs) have improved. However, the identification of new vulnerabilities for this heterogeneous group of cancers remains a priority. Methods: Using two libraries of compounds selected for potential repurposing, we identified the inhibitors of nicotinamide phosphoribosyltransferase (NAMPT) and histone deacetylases (HDAC) as the agents with the highest activity. We validated the hits in an expanded set of neuroendocrine cell lines and examined the mechanisms of action. Results: In Kelly, NH-6, and NCI-H82, which are two neuroblastoma and one small cell lung cancer cell lines, respectively, metabolic studies suggested that cell death following NAMPT inhibition is the result of a reduction in basal oxidative phosphorylation and energy production. NAMPT is the rate-limiting enzyme in the production of NAD+, and in the three cell lines, NAMPT inhibition led to a marked reduction in the ATP and NAD+ levels and the catalytic activity of the citric acid cycle. Moreover, comparative analysis of the mRNA expression in drug-sensitive and -insensitive cell lines found less dependency of the latter on oxidative phosphorylation for their energy requirement. Further, the analysis of HDAC and NAMPT inhibitors administered in combination found marked activity using low sub-lethal concentrations of both agents, suggesting a synergistic effect. Conclusion: These data suggest NAMPT inhibitors alone or in combination with HDAC inhibitors could be particularly effective in the treatment of neuroendocrine neoplasms.

## 1. Introduction

Neuroendocrine neoplasms (NENs) constitute a heterogeneous group of malignancies, arising from the cells of the endocrine and nervous system and classified via the originating site of the primary tumor [1]. They represent less than 2% of all malignancies, with a prevalence in the United States of less than 200,000 cases/year, making NENs an orphan disease [2,3]. However, despite their low prevalence, a majority of NENs are invariably lethal. The vast majority of NENs occur in the gastrointestinal tract, lung, and pancreas [4] and range from indolent to aggressive in their clinical comport, where they can be either hormonally silent or can produce diverse hormones. This heterogeneity has impeded our understanding of the biology of the disease [5].

The primary treatment of patients with localized neuroendocrine neoplasms is surgical, but with metastatic progression, prognosis is poor and therapeutic options are limited [6]. Current treatment options alone or combined with surgery include somatostatin analogues [7], the mTOR inhibitor everolimus [8], the tyrosine kinase inhibitor sunitinib [9], peptide receptor radionuclide therapy (PRRT) [10], and other chemotherapeutics [11], with treatment selection based in part on the aggressiveness of the disease. Recently KPT-9274, an inhibitor of nicotinamide phosphoribosyltransferase (NAMPT) and p21-activated kinase 4 (PAK4), was found active in pancreatic neuroendocrine tumors [12] and is currently being evaluated in a human phase 1 study in patients with advanced solid malignancies. Despite progress, new therapeutics are critically needed for this often-lethal disease.

With a goal of identifying novel drug classes active against NEN, we explored the activity of available, approved, and experimental drugs against NEN cell lines. Using three human neuroendocrine cell lines, NCI-H69, DMS79, and NCI-H72, we screened two small molecule drug libraries selected for potential repurposing, the National Center for Advancing Translational Sciences (NCATS) Pharmaceutical Collection (NPC), and the Mechanism Interrogation PlatE (MIPE) library. Two classes of agents emerged from this screen: inhibitors of nicotinamide phosphoribosyltransferase (NAMPT), and histone deacetylases (HDAC). We demonstrated both classes of inhibitors to be highly active as single agents and synergistic when combined even at low sub-lethal concentrations. Since both previous and current studies have indicated that the inhibitors of both nicotinamide phosphoribosyltransferase (NAMPT), and histone deacetylases (HDAC) have a profound impact on the synthesis of critical tricarboxylic acid cycle (TCA) intermediates, we hypothesized that their use in combination would result in marked metabolic stress that could lead to cell death, and thus decided to explore their activity in combination.

## 2. Material and Methods

### 2.1. Reagents

GMX-1778 and STF-118804 were purchased from APExBIO (Houston, TX, USA). Romidepsin (depsipeptide) was purchased from Sigma-Aldrich (St. Louis, MO, USA). CA3 (CIL56) was purchased from Selleckchem (Houston, TX, USA). pBABE-puro (1764) and pBABE YAP1 (15682) were from Addgene (Watertown, MA, USA).

### 2.2. Cells

The initial screen included three cell lines from the NCI-60; the validation panel included 13 cell lines of neuroendocrine origin, including two of neuroblastoma origin. Neuroblastoma cell lines, whose expression profile resembles pheochromocytoma [13], were included in the cell panel to broaden the representation of neuroendocrine tumor types. Since the CHP-126 cell line grew poorly in the culture, it was substituted with the NH-6 neuroblastoma cell line. Cell lines from the NCI-60 were obtained from the NCI Anticancer Drug Screen; NCI-H727 was purchased from ATCC; and GCIY and NH6 were obtained from Riken Cell Bank (Tsukuba, Japan). Kelly and OE-19 were obtained from Sigma-Aldrich (St. Louis, MO, USA), and NCI-H2342, NCI-H82, and SNU-119 were from Charles River Laboratories (Wilmington, MA, USA). UOK-269 and sublines were kindly provided by Dr. W. Marston Linehan (NCI, Bethesda, MD, USA). The cells were cultured in the appropriate culture medium (Gibco Laboratories, Gaithersburg, MD, USA) and supplemented with 10% fetal bovine serum, 100-units/L penicillin-streptomycin, and 1% glutamine. Cell lines were authenticated and tested for mycoplasma.

Plasmids were transfected into the Kelly cells using the Fugene HD transfection reagent (Promega Corp, Madison, WI, USA) according to the manufacturer’s protocol. At 48 h post-transfection, the cells (1 × 10^6^) were seeded in 100 mm dishes in a complete medium containing 2 μg/mL puromycin. After 2 weeks, the selection-isolated colonies were propagated for further analysis.

### 2.3. Libraries Screened for Active Compounds

Two libraries of compounds that were selected for potential repurposing were screened: (1) the NCATS Pharmaceutical Collection (NPC), a library of ~2750 small molecule drugs approved for use by the USFDA or related agencies in foreign countries, and (2) the NCATS Mechanism Interrogation PlatE (MIPE) library, which includes 1920 mechanistically annotated agents prioritized for clinical relevance.

### 2.4. Metabolic Analysis

The determination of the ATP and NAD+ levels and the metabolic profiling of the Kelly, NCI-H82, and UOK-269 cells were performed by the Metabolomics Core at the AdventHealth Research Institute (Orlando, FL, USA). Briefly, the cells were treated with GMX1778 and STF-118804 at the described concentration for 48 h. The cells were pelleted, frozen, and sent for analysis. The bars represent the standard deviation of three independent experiments. The data were analyzed using GraphPad Prism 8 software (San Diego, CA, USA). *p*-Values were computed using two-way ANOVA.

### 2.5. Western Blotting

Western blotting was carried out according to the manufacturer’s instructions. The membranes were incubated with primary antibodies, SLC52A3 (Thermo Fisher, Waltham, MA, USA), YAP1, PARP, beta-actin (Cell Signaling, Danvers, MA, USA), and secondary antibodies, IRDye^®^ 680RD Goat anti-Mouse and IRDye^®^ 800CW Goat anti-Rabbit (LI-COR Biosciences, Lincoln, NE, USA). The signal was quantitated using the Odyssey CLx Imaging System, and the exposures within the linear range were captured using ImageStudio software V3.1 (LI-COR Biosciences, Lincoln, NE, USA).

### 2.6. Annexin V Assay

The cells (4 × 10^5^) were seeded in 35 mm dishes containing 7 mL of complete medium and exposed to increasing concentrations of GMX-1778 and STF-118804 for 48 hr. Apoptosis was measured as previously described (36) using the annexin V fluorescein isothiocyanate (annexin V-FITC) Apoptosis Detection Kit (BD Biosciences, San Diego, CA, USA) according to the manufacturer’s instructions. Following the treatment, the annexin positive cells were quantitated via FACS analysis (Becton Dickinson, Ranklin Lakes, NJ, USA) using FCS Express Cytometry Software (Tree Star, Inc., Ashland, OR, USA). The percentage of annexin positive cells was calculated and the delta between the treated and untreated control was determined.

### 2.7. Transcriptome Analysis

Total RNA from the NCI-H82, Kelly, NH6, UOK-269, OE-19, NCI-H2342, GCIY, and SNU-119 cell lines was isolated using the RNeasy Mini Kit (QIAGEN, Valencia, CA, USA) according to the manufacturer’s instructions. RNA quality was assessed on the Agilent Bioanalyzer (Agilent Technologies, Santa Clara, CA, USA). RNA libraries were prepared from poly-A pull-down-enriched mRNAs from the total RNA samples (Illumina TruSeq RNA prep kit, San Diego, CA, USA) and were sequenced at the Columbia Genome Center using NovaSeq6000 (Illumina, San Diego, CA, USA). Data were visualized in Qlucore Omics Explorer v. 3.4 (Qlucore AB, Lund, Sweden), including principal component analysis (PCA), heat maps, and unsupervised hierarchical clustering. Gene Set Enrichment Analysis (GSEA) and Ingenuity Pathway analysis (QIAGEN IPA) were used for network- and pathway-focused analyses of transcriptomics data. The expression data are deposited in the Gene Expression Omnibus (GEO) (https://www.ncbi.nlm.nih.gov/geo/, accessed on 1 November 2022) under the accession number GSE216537.

### 2.8. Cytotoxicity Assays

For the initial screen, the CellTiter-Glo assay was used to quantitate the viability of three human neuroendocrine cell lines, NCI-H69, DMS79, and NCI-H727 (Appendix A), seeded at 1000 cells/well in 1536-well plates and exposed to a 10,000-fold range of drug concentrations for 48 h. In the subsequent assays, the cells (5 × 10^3^/well) were seeded in white 96-well plates (Genesee Scientific, El Cajon, CA, USA) in 0.1 mL of complete medium. The cells were exposed to the indicated concentration of romidepsin for 6 h, then washed and grown for 42 h in the presence of increasing concentrations of GMX-1778 and STF-118804. Cell viability was assessed using the CellTiter-Glo assay (Promega Corp, Madison, WI, USA) and the microplate reader PHERAstar FS (BMG LABTECH Inc., Cary, NC, USA). The Excess of Bliss (EOB) score was used for synergy evaluation.

### 2.9. Real-Time ATP Rate Assay

To evaluate the ATP production, a real-time ATP rate assay was performed using a Seahorse XF96 Cell Analyzer according to the manufacturer’s recommendations (Agilent, Santa Clara, CA, USA). Briefly, the cells (Kelly, NH6, OE19 and SNU119) were seeded in Seahorse XF96 cell culture microplates (2.0 × 10^4^ cells per well) in a growth medium supplemented with 10% inactivated fetal bovine serum (FBS). After a 24 h incubation at 37 °C without/with GMX1778 (10 nM) or STF11804 (20 nM), the cells were washed twice with a pre-warmed assay medium (Seahorse XF RPMI medium for the Kelly, OE19, and SNU119 cells and Seahorse XF DMEM medium for the NH6 cells). The oxygen-consumption rate (OCR) and extra-cellular acidification rate (ECAR) values were simultaneously measured following the sequential injections of inhibitors of mitochondrial oxidative phosphorylation: (I) oligomycin (2 µM), an ATP synthase inhibitor; (II) antimycin A (0.5 µM), an inhibitor of complex III; and (III) rotenone (0.5 µM), an inhibitor of complex I, inhibiting uncoupled respiration. The ATP production rates were analyzed using the Agilent Seahorse XF Real-Time ATP Rate Assay Report Generator that automatically calculates the XF Real-Time ATP Rate Assay Parameters (mitoATP Production Rate, glycoATP Production Rate, total ATP Production Rate, XF ATP Rate Index, % glycolysis, and % OXPHOS). The data were normalized to cell numbers by measuring the staining of nuclei (DNA content) using the CyQUANT^®^ Cell Proliferation Assay Kit (Invitrogen, Waltham, MA, USA) following the manufacturer’s recommendations. *p*-Values were computed using a paired *t*-test result analysis for statistical significance.

### 2.10. Statistical Analysis

For the cytotoxicity assays, the data were analyzed using Prism and excel software. All experiments were performed in three biological replicates unless stated otherwise in the relevant section. For the metabolic assays, the data were analyzed using GraphPad Prism 8 software (San Diego, CA, USA). *p*-Values were computed using a two-way ANOVA analysis. Statistical values including mean and standard deviations were computed for all variables/data points. Relations between the test and control variables were determined using the standard *t*-test. All tests were two tailed, and a *p*-value of less than 0.05 was considered statistically significant.

## 3. Results

### 3.1. Identification of Potential Novel Chemotherapeutics for Neuroendocrine Neoplasms

The initial screen for the novel small molecules that could inhibit neuroendocrine cell growth used the NCATS NPC and the MIPE libraries and three human neuroendocrine cell lines, NCI-H69, DMS79, and NCI-H727, and tested 4670 compounds (Appendix A). From this analysis, two classes of agents emerged: the inhibitors of nicotinamide phosphoribosyltransferase (NAMPT), and histone deacetylases (HDAC). These were selected for further validation using an extended neuroendocrine cell line panel with diverse origins and an MYC status (Figure 1 and Table 1, Appendix A). The sensitivity to the HDAC inhibitors was confirmed in 9 of the 13 cell lines while the NAMPT inhibitors showed high cytotoxicity at low concentrations in 10 of the 13 cell lines (Figure 1 and Appendix A).

### 3.2. Sensitivity and Resistance to NAMPT Inhibition

Following the validation in the extended neuroendocrine cell line panel, we next examined the sensitivity to NAMPT inhibition using a panel comprised of different solid tumor cell lines. The sensitivity to either GMX-1778 or STF-118804 was assessed via FACS analysis after a 48 h drug exposure (Figure 2A,B). With IC50 values of 0.9 to 30 nM, the NH-6, Kelly, and NCI-H82 cells were most sensitive to the NAMPT inhibitors in the solid tumor cell line panel, with the remaining cells insensitive even to 50 nM concentrations of the NAMPT inhibitors.

### 3.3. Distinct Molecular Signatures Characterize Sensitivity and Resistance to NAMPT Inhibition

To gain insight into the molecular mechanisms responsible for the differential sensitivity to NAMPT inhibition, we performed RNASeq analysis to examine the gene expression profiles in the solid tumor cell line panel. PCA plot visualization and the unsupervised hierarchical clustering of the untreated cells clearly separated the cells sensitive to NAMPT inhibition (NH-6, Kelly, and NCI-H82) from those shown to be insensitive (SNU-119, OE19, GCIY, NCI-H2342, and UOK-269) (Figure 3A,B). A total of 2618 genes were found to be differentially expressed (DEG) between the two groups of cells (cut-off: S/S_max_ > 0.1, Log2 fold change > 1, *p* value < 0.05). Amongst these, the genes comprising the Broad Institute Hallmark pathways of hypoxia (48/196), glycolysis (36/195), and cholesterol (15/73) were found to be differentially expressed using the Gene Set Enrichment Analysis (Appendix A). The reduced expression of many of the genes in the glycolysis and cholesterol pathways in the sensitive cells raised the possibility of glucose consumption, and its conversion to pyruvate via glycolysis to meet energy requirements could be restricted, rendering the cells more vulnerable to agents that disrupt the TCA cycle (Figure 3C).

Not surprisingly, the genes in the neuroendocrine-related Pancreas Beta Hallmark pathway were over-expressed in the sensitive cells. Similarly, we also found that 557 genes annotated as neuroendocrine via the Ingenuity Pathway Analysis (IPA) clearly separated the two groups of cells, which was also shown in the genes encoding proteins expressed in the adrenal medulla as annotated in The Human Protein Atlas (Appendix A).

### 3.4. NAMPT Inhibitor Effect on ATP and NAD Synthesis in Sensitive and Resistant Cell Lines

The preceding experiments demonstrated that the Kelly and NH6 neuroblastoma cell lines and the NCI-H82 small cell lung cancer cell line were most sensitive to the inhibition of NAMPT, the rate limiting enzyme of the NAD+ salvage pathway [14] (Figure 1 and Figure 2). NAD+ depletion following the NAMPT inhibition has been reported to lead to ATP exhaustion and cell death [15,16]. To assess the metabolic effect of NAD^+^-depletion, we assessed ATP production and the level of TCA intermediates via LC-MS/MS and Seahorse analyses in the cells with a range of sensitivity to the NAMPT inhibitors, GMX-1778 and STF-118804. Marked reductions in the intracellular ATP and NAD+ levels were observed in the Kelly cells after 48 h of exposure to 2 nM GMX-1778 and 10 nM STF-118804 and in NCI-H82 to 30 nM of either drug (Figure 4A). Metabolomic studies have also suggested that the reduced availability of tricarboxylic acid cycle (TCA) intermediates can induce metabolic cell stress and cell death [16]. When the levels of TCA intermediates were evaluated in the presence of the two NAMPT inhibitors, reductions were observed in Kelly, NCI-H82, and the UOK-269 cell lines (Figure 4B,C). Additionally, we conducted Seahorse metabolic flux analyses to determine the impact of NAMPT inhibition on the oxygen consumption rate (OCR) and extracellular acidification (ECAR) in NAMPT-sensitive (Kelly, NH6) and -insensitive (SNU-119, OE-19) cell lines. OCR and ECAR are indicators of mitochondrial respiration and glycolysis, respectively, and their levels of ATP production. As shown in Figure 4D, ATP production in the NAMPT-sensitive Kelly and NH6 cells occurs preferentially via mitochondrial respiration (MitoATP), while in the NAMPT-insensitive SNU-119 and OE-19 cells, mitochondrial respiration and glycolysis contribute comparably to ATP production. NAMPT inhibition led to a greater decrease in ATP production in the NAMPT-sensitive (20 to 50%) than in the insensitive (10 to 20%) cells (Figure 4E), suggesting sensitivity to NAMPT inhibition could be explained via the reliance of the sensitive cells on mitochondrial respiration as their primary source of ATP.

Seeking additional clues regarding the sensitivity of the neuroendocrine cells to NAMPT inhibition, we re-examined the basal RNA expression levels and found SLC52A3 (but not SLC52A1 or SLC52A2), and the Yes-Associated Protein 1 (YAP1) amongst the genes expressed at lower levels in the drug-sensitive group (Appendix A). SLC52A family members encode proteins involved in riboflavin (vitamin B2) transport and a Western blot analysis of the SLC52A3 protein in the cell line panel confirmed the lower levels of expression in the NH6 and Kelly cells corresponding with the RNAseq analysis (Appendix A). Based on this reduced SLC52A3 expression and previous studies showing a protective role of riboflavin [17,18], we investigated if riboflavin treatment would protect the NH6 and Kelly cells following the NAMPT inhibition. As shown in Appendix A, the addition of riboflavin at final concentrations of 25 and 50 nM decreased cell death following the exposure to the NAMPT inhibitors in both the NH6 and Kelly cells, although not in the NCI-H82 cells. This experiment suggests a possible contribution of the lower protein expression of SLC52A3 in the NH6 and Kelly cells to their increased the sensitivity to the NAMPT inhibitors.

We then turned to examine YAP1, a transcriptional co-activator [19] that associates with the transcriptional enhancer domain (TEAD) family members, inducing the expression of genes involved in cell proliferation and the inhibition of apoptosis [20,21]. Protein levels of YAP1, whose mRNA like that of SLC52A3 is expressed at lower levels in the NAMPT-sensitive cells, were also found to be lower in the NAMPT-sensitive than the insensitive cells (Figure 5A). A lower YAP1 expression was also observed in the NAMPT-sensitive cells of the panel, which is used to confirm the results of the primary screening (Figure 5B).

Given the reduced expression and previous studies suggesting a role for YAP1 in the inhibition of apoptosis [22] and drug resistance [23], we proceeded to investigate if the ectopic expression of YAP1 in the Kelly cells (Figure 5C) could reduce the sensitivity to NAMPT inhibition. The FACS analysis of the Kelly cells transfected with an empty vector (Kelly/Babe) or containing the YAP1 gene (Kelly/YAP1) found the Kelly/YAP1 cells to be less sensitive than the Kelly/Babe control to 48 h exposure of 1 and 2 nM of GMX-1778 (Figure 5D). Interestingly, in the Seahorse analyses when compared to Kelly and Kelly/Babe, the less sensitive Kelly/YAP1 cells showed increased ATP production via glycolysis (Figure 5E). We next investigated the effect of inhibition of the YAP1 function in the resistant GCIY cell line using CA3 (CIL56), a known inhibitor of YAP1/TEAD levels/transcriptional activity [24]. Treatment of the GCIY cells with CA3 and GMX-1778 demonstrated a clear synergistic effect of the two drugs in combination. Cell viability, assessed via FACS analysis, was reduced to 40% in the presence of the two drugs compared to either GMX-1778 (80–90%) or CA3 (70%) alone (Figure 5F). The expression analysis of YAP1 and PARP-1 in the GCIY cells treated for 48 h with GMX-1778 and CA3 demonstrated the inhibition of YAP1 with the two-drug combination (Figure 5G). Furthermore, an increase in PARP-1 cleavage was observed in the combination treatment with respect to CA3 and GMX-1778 alone (Figure 6B), suggesting a correlation between the insensitivity to NAMPT inhibition and YAP1 expression. Interestingly, the seahorse analyses in SNU-119 and OE-19 treated with GMX-1778 and STF-118804 plus CA3 showed reduced ATP production at both the OCR and ECAR levels (Appendix A), confirming the efficacy of the combination at the metabolic level.

### 3.5. The Combination of HDAC and NAMPT Inhibitors Is Synergistic in Neuroendocrine Cells

Previous work in our laboratory has demonstrated that treatment of the neuroendocrine cancer cells with romidepsin for six hours can reduce the basal OCR by 25–40% and reduce acetyl-CoA levels with subsequent cell death. This reduction in the OCR and acetyl-CoA levels can be rescued via exogenous citrate, acetate, and acetyl-CoA [25]. Considering the effect of NAMPT inhibition on the levels of TCA cycle intermediates, including citrate and acetyl-CoA, and the sensitivity of the expanded neuroendocrine cell panel to the HDAC and NAMPT inhibitors, we evaluated a potential synergistic interaction between the two class of inhibitors.

We also theorized that HDAC inhibition could suppress the YAP expression, as previously reported [26]. Treatment of the NAMPT-insensitive OE-19, SNU-119, and GCIY cells with increasing concentrations of romidepsin reduced the YAP1 expression (Figure 6A). Furthermore, treating the NAMPT-insensitive cells with romidepsin in combination with GMX-1778 resulted in a decrease in YAP1 expression and a concomitant increase in PARP-1 cleavage (Figure 6B). The immunoblot analysis of YAP1 and cleaved PARP-1 in the GCIY cells treated for 48 h with both romidepsin and GMX-1778 emulated the results obtained with the combination of CA3 and GMX-1778, suggesting a combination of HDAC and NAMPT inhibitors is effective in increasing drug sensitivity.

Finally, when the NH-6, NCI-H82, and Kelly cells were exposed for 6 h to romidepsin and then to the increasing concentrations of GMX-1778 or STF-118804 for 48 h, clear synergy was observed for both the two-drug combinations even with the combination of sub-lethal concentrations of the HDAC and NAMPT inhibitors (Figure 6C).

## 4. Discussion

Neuroendocrine neoplasms (NEN) constitute a diverse group of malignancies whose heterogeneity has hindered the identification of effective clinical therapies. Therefore, the discovery of vulnerabilities in NEN and the small molecules targeting them remains a priority. In this study, we have identified two classes of inhibitors, those targeting nicotinamide phosphoribosyltransferase (NAMPT) and histone deacetylases (HDAC), with NAMPT inhibitors as a drug class being particularly active against some neuroendocrine cells. We identify a preferential dependence of the NEN cells on oxidative phosphorylation for energy, and the reduced expression of both SLC52A3, a riboflavin protein transporter, and the Yes-Associated Protein, YAP1, a core component of the Hippo pathway involved in the regulation of metabolism, promoting glycolysis, and glutaminolysis, as identifiable factors that could guide treatment decisions in a precision oncology strategy. Finally, we provide compelling evidence of synergy when both the NAMPT and HDAC inhibitors are used together in what would be a novel therapeutic strategy.

The screening of two libraries of compounds using three neuroendocrine tumor derived cell lines identified NAMPT and HDAC inhibitors as two classes of compounds particularly effective in reducing neuroendocrine tumor cell viability and proliferation. In this study, we have investigated the effect of NAMPT and HDAC inhibitors on energy metabolism, ATP production, and ultimately cell death. From the initial screen and its subsequent validation/confirmation, neuroendocrine neoplasms were identified as sensitive to the effect of the NAMPT inhibitors (GMX-1178 and STF-118804). When two representative cell lines, Kelly (neuroblastoma) and NCI-H82 (small cell lung cancer) were exposed to GMX-1778 and STF-118804, a decrease in the ATP and NAD+ levels were observed when compared to the levels in the untreated cells. NADH, the reduced form of NAD, drives the generation of ATP via oxidative phosphorylation (OXPHOS). As an essential coenzyme, NAD gains two electrons and a proton from the substrates at multiple steps of the TCA cycle, being consequently reduced to NADH. The TCA catalytic activity analysis in the two sensitive cell lines, together with the gene expression profiling analysis and the ATP production analysis between the NAMPT-sensitive and -insensitive cells, suggest that drug insensitivity in the latter can be attributed at least in part to their ability to leverage both mitochondrial respiration and glycolysis for the energy requirement, while the sensitive cell lines derive their energy primarily from mitochondrial respiration and undergo oxidative stress after NAMPT inhibition. Indeed, the increased expression of genes involved in hypoxia and glycolysis in the NAMPT-insensitive cell lines supports this hypothesis.

Riboflavin, known as vitamin B2, is involved in a variety of metabolic pathways and performs key metabolic functions by mediating the transfer of electrons in a biological oxidation–reduction reaction [27]. Riboflavin is the precursor of the coenzymes, flavin mononucleotide (FMN) and flavin adenine dinucleotide (FAD), which are pivotal in the transfer of electrons in biological oxidation–reduction reactions [28,29]. In the TCA cycle, FAD, the more complex and abundant form of flavin, is reduced to FADH2 during the oxidation of succinate to fumarate. When FADH2 reverts to FAD, its two high-energy electrons are conveyed to the electron transport chain to produce ATP [30]. Members of the SLC52A family encode the proteins involved in riboflavin (vitamin B2) transport and we reasoned that the lower flavin levels could render the cells more vulnerable to the disruption of the TCA cycle and that this could be exacerbated by the addition of NAMPT inhibitors. Our data suggest that the reduced expression of both SLC52A3, a riboflavin protein transporter, and YAP1, a core component of the Hippo pathway involved in the regulation of metabolism, promoting glycolysis, and glutaminolysis [31], contributes to the sensitivity of the Kelly, NH6, and NCI-H82 cells to the NAMPT inhibitors. Moreover, reducing the YAP1 expression/activity either with CA3 (CIL56), a known inhibitor of YAP1/TEAD levels/transcriptional activity or with the histone deacetylase inhibitor, romidepsin, increased the sensitivity to the NAMPT inhibitors in the drug-insensitive GCIY, OE-19. and SNU-119 cells. The evidence that a reduced expression of SLC52A3, a riboflavin transporter protein, and YAP1, a core component of the Hippo pathway, play roles in the sensitivity of the Kelly and NH6 cells to the NAMPT inhibitors also supports the oxidative stress hypothesis.

Finally, the epigenetic and metabolic vulnerabilities in the neuroblastoma and small cell lung cancer cell lines serving as representative NENs were evident in the drug–drug combination analysis where the combination of HDAC and NAMPT inhibitors demonstrated synergy even when both agents were combined at low sub-lethal concentrations. HDAC inhibitors, including five approved by regulatory agencies worldwide with excellent toxicity profiles, are both unique in their mechanism of action and not fully explored in rational combinations. Our data demonstrating the synergy in leading to oxidative stress and cellular toxicity at sub-lethal concentrations with comparably low concentrations of NAMPT inhibitors could, in combination, reduce the amount of NAMPT inhibitor needed, improving their tolerability.

Gradually, the intersection of metabolism and epigenetics has become manifest in oncology. The state of nutrient availability has been shown to impact the state of differentiation, gene expression, and alterations in the epigenome [32]. Thus, the notion of exploiting a metabolic vulnerability in the neuroendocrine cells through combining the inhibitors of NAMPT and HDACs offers an untapped opportunity for a new therapeutic strategy. Two decades of studies suggest that HDAC inhibition could be important in neuroendocrine neoplasms, from the observation of a patient with a pancreatic islet cell tumor with a response to romidepsin in a Phase I study, activity in small clinical trials, through numerous in vitro studies, and more recently, in key bioinformatics analyses [33,34,35,36,37,38]. However, single agent therapy is likely to be insufficient [36,37], pointing to the need for rational combinations. Reducing the YAP1 expression, along with a double hit on the TCA cycle—depleting the essential molecule acetyl CoA via HDAC inhibition and blocking key intermediate steps dependent on NAD cofactors via NAMPT inhibition could be highly effective in neuroendocrine neoplasms and should be pursued with a goal of developing translational clinical trials.

## 5. Conclusions

In conclusion, our results have identified the simultaneous inhibition of HDACs and NAMPT as a novel approach for the treatment of neuroendocrine neoplasms. The role played by YAP1 in the sensitivity to the NAMPT inhibitors and its reduction via romidepsin aggravates the depletion of NAD+ and the reduction in the ATP production that occurs with the inhibition of NAMPT, leading to enhanced cellular cytotoxicity. Combinations of HDAC and metabolic inhibitors should be evaluated with the aim of identifying new therapeutic options for the treatment of neuroendocrine neoplasms.

## Figures and Tables

**Figure 1 cancers-15-04960-f001:**
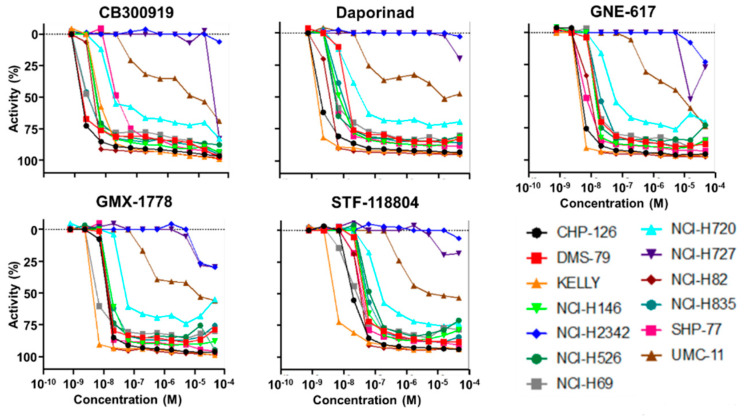
Confirmation of activity of NAMPT inhibitors on neuroendocrine cancer cells. The sensitivity to the NAMPT inhibitors, CB300919, Daporinad, GNE-617, GMX-1778, and STF-118804, was confirmed using a panel of 13 neuroendocrine cell lines. Activity is plotted as percent of control. Concentrations are Log [M].

**Figure 2 cancers-15-04960-f002:**
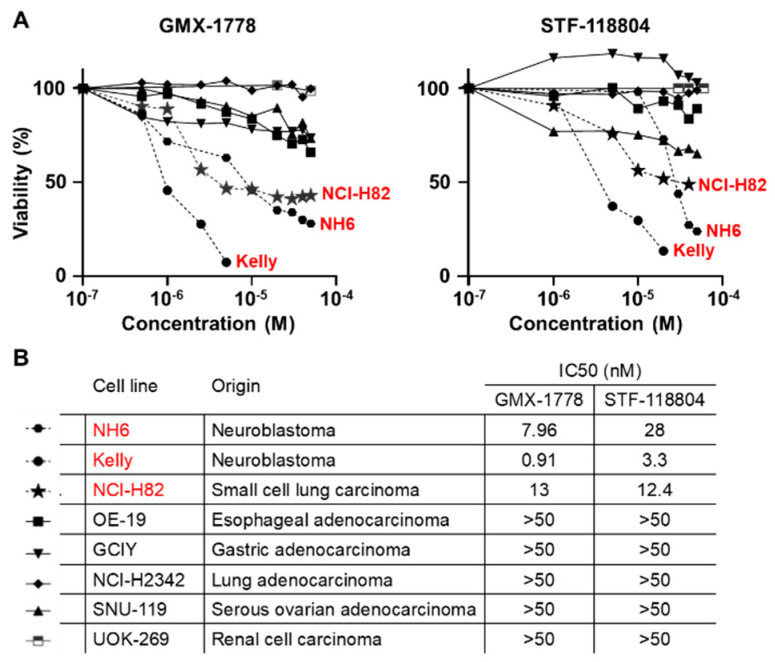
(**A**) Sensitivity to GMX-1778 and STF-118804. FACS analysis of cells after exposure to increased concentration of GMX-1778 and STF-118804 for 48 h. Cell viability is measured as percent of control. Concentrations are Log [M]. (**B**) Cell panel showing origin of each cell line and IC50 values. IC50 values were calculated using GraphPad Prism software as average of three independent experiments.

**Figure 3 cancers-15-04960-f003:**
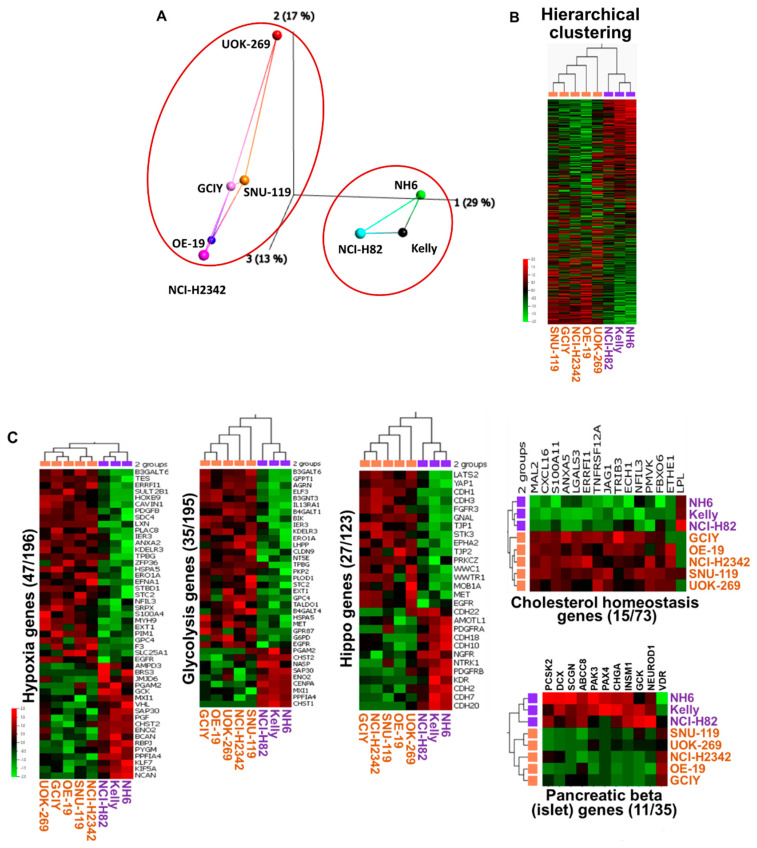
Differential expression analysis of NAMPT inhibitors sensitive and resistant cell lines. (**A**) Principal component analysis (PCA) shows a clear distinction based on cell type. (**B**) Unsupervised hierarchical clustering divided the samples according to cell type (Var > 0.3, Log2-transformed. 0.01 threshold). Unsupervised hierarchical clustering based on the 2618 most variable (Var > 0.5) genes divided the samples according to cell type. (**C**) Differentially expressed genes in the Broad Institute’s Hallmark Pathways between sensitive and insensitive cells are shown. Hypoxia pathway, 48/196; Glycolysis pathway, 36/195; Cholesterol pathway, 15/73; and Pancreatic beta genes, 11/35. *p*-Value < 0.05, Log2 foldchange > 1. While not among the 50 Hallmark gene sets, we also noted differentially expressed genes in the Hippo pathway (28/123).

**Figure 4 cancers-15-04960-f004:**
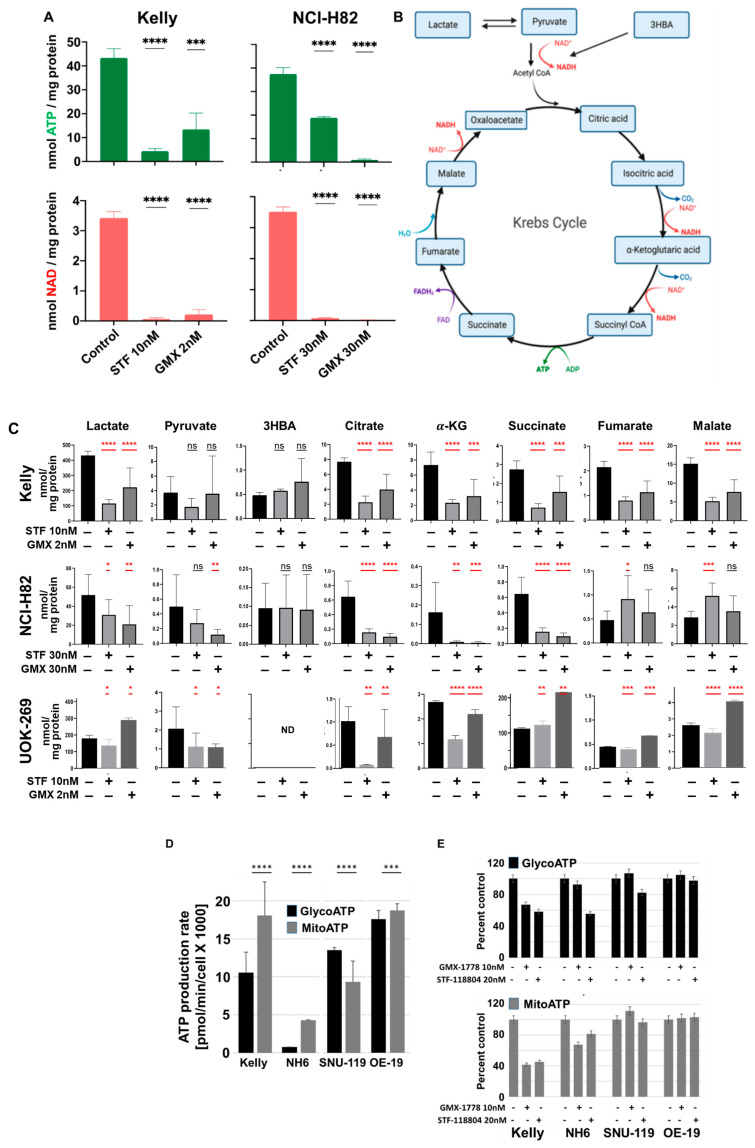
GMX-1778 and STF-118804 reduce energy production in Kelly and NCI-H82. (**A**) ATP and NAD+ levels in Kelly and NCI-H82 cells after 48 h of exposure to GMX-1778 and STF-118804. (**B**) Schematic representation of the cycle of tricarboxylic acids (TCA). Mitochondrial reactions and enzymes constituting the TCA cycle are summarized. (**C**) Measure of TCA cycle intermediates in Kelly, NCI-H82, and UOK-269 cells after 48 h of exposure to GMX-1778 and STF-118804. ATP, NAD+, and Intermediate levels are expressed as ng/mg of protein. Bars represent standard deviations. Calculated *p* value 0.01 < *, 0.001 < **, 0.0005 < *** and 0.0001 < **** are shown. (**D**) Levels of ATP production via mitochondrial respiration and glycolysis in Kelly, NH6, SNU-119, and OE-19 cell lines. Effect of NAMPT inhibition on ATP production. Cells were exposed to GMX-1778 (10 nM) and STF-118804 (20 nM) for 24 h. ATP production is expressed as pmol/min/cell × 1000. ATP production in (**E**) is expressed as 100% in untreated cells (control). Calculated *p* value, 0.0001 < ****, 0.01 < ***. Bars represent standard deviations of three independent results.

**Figure 5 cancers-15-04960-f005:**
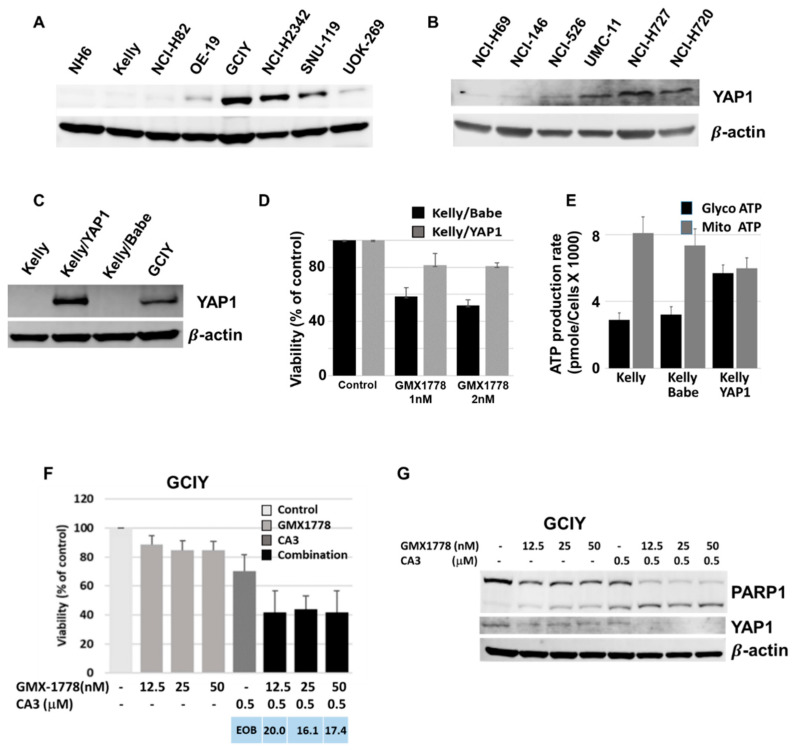
YAP1 deficiency plays a role in sensitivity to GMX-1778. (**A**,**B**) YAP1 protein expression in sensitive and resistant cell lines. (**C**) Ectopic YAP1 protein expression in Kelly cells transfected with empty (Kelly/Babe) or YAP1 (Kelly/YAP1) expressing vector. (**D**) FACS analysis of Kelly/Babe and Kelly/YAP1 cells exposed to GMX-1778 for 48 h. Cell viability is expressed as percent of control. (**E**) Levels of ATP production via mitochondrial respiration and glycolysis in Kelly, Kelly/Babe, and Kelly/YAP1 cell lines. ATP production is expressed as pmol/min/cell × 1000. (**F**) FACS analysis of GCIY cells exposed for 48 h to GMX-1778, CA3, and their combination. Cell viability is expressed as percent of control. Excess of bliss (EOB) values represent average measurements of synergy of three independent experiments. (**G**) PARP1 and YAP1 protein expression in GCIY cells exposed for 48 h to GMX-1778, CA3, and their combination. Bars represent standard deviations of three independent results. The uncropped bolts are shown in Appendix A.

**Figure 6 cancers-15-04960-f006:**
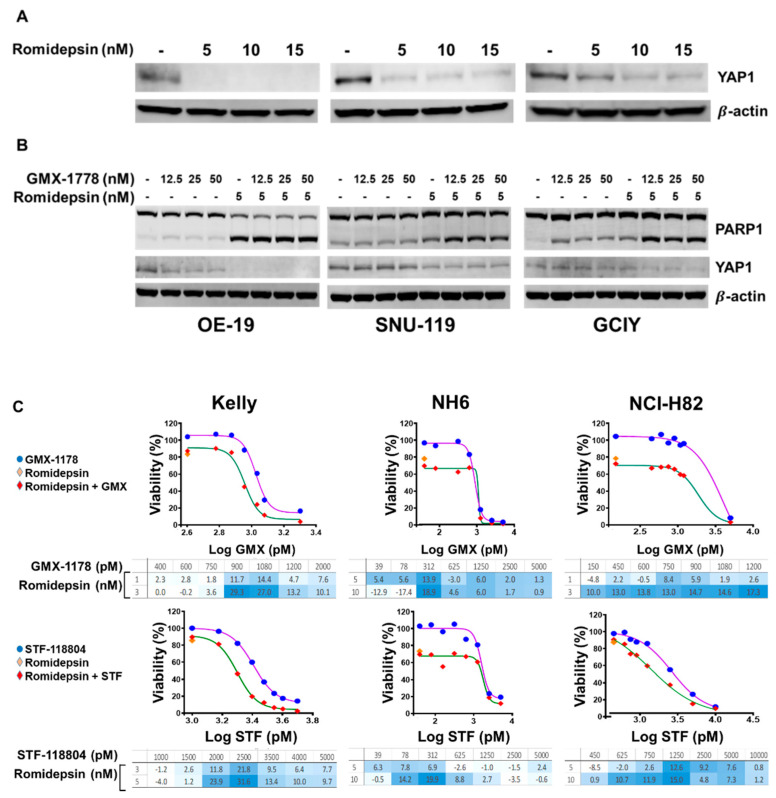
Synergistic effect of NAMPTs and romidepsin combination in resistant and sensitive cells. (**A**) YAP1 protein expression in OE-19, SNU-119, and GCIY cells exposed for 48 h to increased concentration of romidepsin. (**B**) PARP and YAP1 protein expression in OE-19, SNU-119, and GCIY exposed for 48 h to GMX-1778, romidepsin, and their combination. (**C**) Kelly, NH6, and NCI-H82 cells were exposed 6 h to romidepsin, medium was substituted, and cells grown for 42 h in presence of increasing concentrations of GMX-1778 and STF-118804. Viability, expressed as percent of control, was determined at 48 h using Cell-Titer Glo and is the average of three independent experiments. Excess of bliss (EOB) values represent average measurements of synergy of three independent experiments. The uncropped bolts are shown in Appendix A.

**Table 1 cancers-15-04960-t001:** Activity concentration values of NAMPT inhibitors. Activity concentration (AC50) values (μM) of each NAMPT inhibitor relative to the 13 neuroendocrine cell lines. Also shown is the tumor origin of each cell line, and MYC amplification status, including one cell line with MYC circular RNA.

Cell Type	CB300919	GNE-617	STF-118804	GMX-1778	FK866	Origin	MYC
	AC50 (mM)						
CHP-126	0.002	0.005	0.019	0.013	0.002	Neuroblastoma	N-MYC
DMS-79	0.002	0.015	0.042	0.013	0.012	Small cell lung cancer	-
KELLY	0.006	0.004	0.005	0.004	0.001	Neuroblastoma	N-MYC
NCI-H146	0.005	0.013	0.047	0.017	0.006	Small cell lung cancer	N-MYC
NCI-H526	0.005	0.013	0.052	0.015	0.005	Small cell lung cancer	N-MYC
NCI-H69	0.002	0.015	0.017	0.005	0.006	Small cell lung cancer	N-MYC
NCI-H720	0.013	0.037	0.105	0.037	0.017	Lung carcinoid	N-MYC
NCI-H727	37.139	11.744	null	6.604	null	Lung carcinoid	-
NCI-H82	0.004	0.008	0.03	0.012	0.003	Small cell lung cancer	C-MYC
NCI-H835	0.002	0.019	0.052	0.017	0.007	Lung carcinoid	-
SHP-77	0.021	0.006	0.033	0.013	0.005	Small cell lung cancer	MYC circRNA
UMC-11	0.933	2.343	0.66	0.295	0.052	Lung carcinoid	-
NCI-H2342	null	null	null	10.467	null	Lung adenocarcinoma	N-MYC

## Data Availability

The expression data are deposited in the Gene Expression Omnibus (GEO; https://www.ncbi.nlm.nih.gov/geo/) under the accession number GSE216537. Metabolomic data are available on request.

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
