# Peer review of "Novel Therapeutic Strategies Exploiting the Unique Properties of Neuroendocrine Neoplasms"

_cancers, 2023, doi:10.3390/cancers15204960_

Round 1

Reviewer 1 Report

In Maryam Safari’s paper, they discovered a novel approach to neuroendocrine tumors. They re-evaluated two drugs, NAMPT and HDAC, by searching compound databases. Subsequently, through cell line experiments and high-throughput sequencing analysis, they compared the biological mechanisms between drug-sensitive and drug-insensitive cells. They found that these two agents exhibit significant synergistic effects. Overall, Maryam Safari's research is quite intriguing and reads smoothly. However, I have a few suggestions that may help enhance the quality of the article.

1.     For neuroblastoma, MYCN amplification is a crucial pathogenic factor, and cell lines with or without amplification exhibit significant differences. The cell lines used in the article, including Kelly, NH-6, and CHP-126, all have MYCN amplification. Would it be possible to consider adding cell lines without MYCN amplification?

2.     In terms of Log2 fold change > 2, p-value < 0.05, q < 0.35, Should it be the absolute value of log2 fold change >2? In addition, Why did the authors choose q < 0.35?

3.     While many differentially expressed genes are related to the hypoxia, glycolysis, hippo, and cholesterol pathways, Figure 3c shows that upregulated and downregulated genes coexist. Therefore, utilizing methods like GSEA or GSVA may better support the subsequent conclusion of “greater reliance in the drug insensitive cells on glucose”

Author Response

Reviewer #1

Comments and Suggestions for Authors

In Maryam Safari’s paper, they discovered a novel approach to neuroendocrine tumors. They re-evaluated two drugs, NAMPT and HDAC, by searching compound databases. Subsequently, through cell line experiments and high-throughput sequencing analysis, they compared the biological mechanisms between drug-sensitive and drug-insensitive cells. They found that these two agents exhibit significant synergistic effects. Overall, Maryam Safari's research is quite intriguing and reads smoothly. However, I have a few suggestions that may help enhance the quality of the article. 

  1. For neuroblastoma, MYCN amplification is a crucial pathogenic factor, and cell lines with or without amplification exhibit significant differences. The cell lines used in the article, including Kelly, NH-6, and CHP-126, all have MYCN amplification. Would it be possible to consider adding cell lines without MYCN amplification?

Our validation cell line set included 12 neuroendocrine cell lines of which 2 were neuroblastoma. Of the 12 in the validation set, 4 do not have MYCN amplification. The two most sensitive lines of this neuroendocrine set were of neuroblastoma origin, which we included as a phenocopy for pheochromocytoma.  We later substituted NH-6 for CHP-126, due to the latter’s poor growth characteristics. We have updated Table 1 to include MYC information, and we have added text to the manuscript to be sure this is clear.  New Table 1 to be added to manuscript is shown below.

Table 1: Activity Concentrations of NAMPT Inhibitors

Cell Type

CB300919

GNE-617

STF-118804

GMX-1778

FK866

Origin

MYC

AC50 (mM)

CHP-126

0.002

0.005

0.019

0.013

0.002

Neuroblastoma

N-MYC

DMS-79

0.002

0.015

0.042

0.013

0.012

Small cell lung cancer

-

KELLY

0.006

0.004

0.005

0.004

0.001

Neuroblastoma

N-MYC

NCI-H146

0.005

0.013

0.047

0.017

0.006

Small cell lung cancer

N-MYC

NCI-H526

0.005

0.013

0.052

0.015

0.005

Small cell lung cancer

N-MYC

NCI-H69

0.002

0.015

0.017

0.005

0.006

Small cell lung cancer

N-MYC

NCI-H720

0.013

0.037

0.105

0.037

0.017

Lung carcinoid

N-MYC

NCI-H727

37.139

11.744

null

6.604

null

Lung carcinoid

-

NCI-H82

0.004

0.008

0.03

0.012

0.003

Small cell lung cancer

C-MYC

NCI-H835

0.002

0.019

0.052

0.017

0.007

Lung carcinoid

-

SHP-77

0.021

0.006

0.033

0.013

0.005

Small cell lung cancer

Circular MYC RNA

UMC-11

0.933

2.343

0.66

0.295

0.052

Lung carcinoid

-

NCI-H2342

null

null

null

10.467

null

Lung adenocarcinoma

N-MYC

  1. In terms of Log2 fold change >2, p-value < 0.05, q < 0.35, Should it be the absolute value of log2 fold change >2? In addition, why did the authors choose q < 0.35?

The reviewer is correct. The correct term should be log2 fold change >1.
As regards the q value, the Benjamini-Hochberg method was used to calculate this for each gene, and the maximum q-value reached for a given contrast was reported (for p<0.05). The q value was not used as a cut-off and has been deleted from the sentence listing cut-off criteria.

  1. While many differentially expressed genes are related to the hypoxia, glycolysis, Hippo, and cholesterol pathways, Figure 3c shows that upregulated and downregulated genes coexist. Therefore, utilizing methods like GSEA or GSVA may better support the subsequent conclusion of “greater reliance in the drug insensitive cells on glucose”.

Thank you for this insightful comment. It is correct that the directionality of the DEGs is crucial for the interpretation of “greater reliance” of insensitive cells. Indeed, we used Gene Set Enrichment Analysis (GSEA) to evaluate the Hallmarks of cancer included in the manuscript.  We have revised the section accordingly, and also included the Hallmark analysis as Supplementary Table 2. We added the following to the manuscript:

“Amongst these, genes comprising the Broad Institute Hallmark canonical pathways of hypoxia (48/196), glycolysis (36/195), Hippo (28/123) and cholesterol (15/73) were found to be differentially expressed using Gene Set Enrichment Analysis (Supplementary Table 2). Reduced expression of many of the genes in the glycolysis and cholesterol pathways in the sensitive cells, raised the possibility glucose consumption and its conversion to pyruvate via glycolysis to meet energy requirements could be restricted, rendering cells more vulnerable to agents that disrupt the TCA cycle (Figure 3C).

Reviewer 2 Report

1. Please use the term NEN and not NET according to the latest WHO classification. 

2. There is not much information on the statistics. Please make a section dedicated to description of the statistical analysis. 

3. Why was neuroblastoma and not GI NEN cells chosen?

4. The Results paragraph should be re-written. There is much information which should be either in the introduction or the discussion.

5. Why did you chose the specific medical drugs, please make the introduction more clear. 

Author Response

Reviewer #2

Comments and Suggestions for Authors

  1. Please use the term NEN and not NET according to the latest WHO classification. 

We agree with this very practical suggestion and as requested the term neuroendocrine tumor (NET) have been changed to neuroendocrine neoplasm (NEN) throughout the manuscript.

  1. There is not much information on the statistics. Please make a section dedicated to description of the statistical analysis.

A statistical analysis paragraph has been added to the end of the Methods section.

  1. Why was neuroblastoma and not GI NEN cells chosen?

Unfortunately, as the reviewer likely knows, the choice of neuroendocrine cell lines is not as robust as one would like. We decided on the 12 validation cell lines from a larger panel, they were ones that were readily adapted to the high-throughput screen at the NCI, were easy to grow and seemed to provide reproducible results. Note, we did not omit any cell lines because of discordance with any results.

4. The Results paragraph should be re-written. There is much information which should be either in the introduction or the discussion.

This is a very valid point often overlooked as one writes a manuscript. As suggested, we have moved several parts of the Results to the Introduction, Methods, or Discussion.

  1. Why did you choose the specific medical drugs, please make the introduction clearer. 

We wanted to choose drugs that were likely to be given to patients. In the case of romidepsin, while there are several HDAC inhibitors, our laboratory has great experience with this drug, and it is clinically the most potent HDAC inhibitor, and this guided its selection. As regard NAMPT inhibitors none are approved, and we selected instead agents that have been/are in development.

Reviewer 3 Report

This was a well conducted pre-clinical analysis of repurposed therapeutics in multiple neuroendocrine cell lines. 

A few questions:

- at the end of the background, a summary of the current work is provided instead of the hypothesis posed. I would reconsider, unless prior work has been conducted and therefore should be mentioned or cited as the impetus for this project.

- Although in the background you mention the heterogeneity of the neuroendocrine neoplasms, the cell lines used do not cover all types - specifically GI neuroendocrine neoplasms. Is there a reason for that? Is there a reason a lung adenocarcinoma cell line was included? 

- section 3.2 in results. was this cohort of all tumor types used as a control? what was the intention? 

Author Response

Reviewer #3

Comments and Suggestions for Authors

This was a well conducted pre-clinical analysis of repurposed therapeutics in multiple neuroendocrine cell lines. 

A few questions:

  1. At the end of the background, a summary of the current work is provided instead of the hypothesis posed. I would reconsider, unless prior work has been conducted and therefore should be mentioned or cited as the impetus for this project.

We agree with this helpful suggestion that a mention of the hypothesis is valuable and have now added the following to the manuscript at the end of the Introduction:

“Because both previous and current studies have indicated inhibitors of both nicotinamide phosphoribosyltransferase (NAMPT), and histone deacetylases (HDAC) have a profound impact on the synthesis of critical tricarboxylic acid cycle (TCA) intermediates, we hypothesized that their use in combination would result in marked metabolic stress that could lead to cell death, and thus decided to explore their activity in combination.”

  1. Although in the background you mention the heterogeneity of the neuroendocrine neoplasms, the cell lines used do not cover all types - specifically GI neuroendocrine neoplasms. Is there a reason for that? Is there a reason a lung adenocarcinoma cell line was included? 

As noted in the response to Reviewer #2, the choice of neuroendocrine cell lines for the validation panel is not as robust as one would like. We decided on these cell lines from a larger panel, and they were ones that were well known in the research community and were easy to grow and provide reproducible results. The cell lines were selected in part by the NCATS program for use in their high-throughput assays. The lung adenocarcinoma was included as a control in the neuroendocrine validation series shown in Figure 1.

  1. Section 3.2 in results. Was this cohort of all tumor types used as a control? What was the intention? 

We are not fully clear what “section 3.2” refers to, but we think it refers to Figure 2 on page 32. To seek potential explanations for the sensitivity of the neuroendocrine cells, we chose three of the more sensitive neuroendocrine cell lines (Kelly, NCI-H82 and NH6) from the validation panel (later replacing CHP126 with NH6 due to growth characteristics), and then selected 5 additional solid tumor cell lines to form a solid tumor panel to examine sensitivity and resistance. We have clarified this in the text by referring to this as a solid tumor panel.

Round 2

Reviewer 2 Report

Thank you for the improved manuscript. 

I have no further comments.